# A Pilot Study on Proteomic Predictors of Mortality in Stable COPD

**DOI:** 10.3390/cells13161351

**Published:** 2024-08-14

**Authors:** Cesar Jessé Enríquez-Rodríguez, Carme Casadevall, Rosa Faner, Sergi Pascual-Guardia, Ady Castro-Acosta, José Luis López-Campos, Germán Peces-Barba, Luis Seijo, Oswaldo Antonio Caguana-Vélez, Eduard Monsó, Diego Rodríguez-Chiaradia, Esther Barreiro, Borja G. Cosío, Alvar Agustí, Joaquim Gea, on behalf of the BIOMEPOC Group

**Affiliations:** 1Hospital del Mar Research Institute, Respiratory Medicine Department, Hospital del Mar. Medicine and Life Sciences Department, Universitat Pompeu Fabra (UPF), BRN, 08018 Barcelona, Spain; carme.casadevall@upf.edu (C.C.); spascual@psmar.cat (S.P.-G.); ocaguana@psmar.cat (O.A.C.-V.); ebarreiro@researchmar.net (E.B.); 2Centro de Investigación Biomédica en Red, Área de Enfermedades Respiratorias (CIBERES), Instituto de Salud Carlos III, 28029 Madrid, Spain; rfaner@recerca.clinic.cat (R.F.); ady@12o.es (A.C.-A.); lcampos@separ.es (J.L.L.-C.); gpecesbarba@gmail.com (G.P.-B.); lseijo@unav.es (L.S.); eduardmonsomolas@gmail.com (E.M.); borja.cosio@ssib.es (B.G.C.); aagusti@clinic.cat (A.A.); 3Servei de Pneumologia (Institut Clínic de Respiratori), Hospital Clínic—Fundació Clínic per la Recerca Biomèdica, Universitat de Barcelona, 08907 Barcelona, Spain; 4Respiratory Medicine Department, Hospital 12 de Octubre, 28041 Madrid, Spain; 5Unidad Médico-Quirúrgica de Enfermedades Respiratorias, Hospital Universitario Virgen del Rocío, Universidad de Sevilla, 41012 Sevilla, Spain; 6Respiratory Medicine Department, Fundación Jiménez Díaz, Universidad Autónoma de Madrid, 28049 Madrid, Spain; 7Respiratory Medicine Department, Clínica Universidad de Navarra, 31008 Madrid, Spain; 8Institut d’Investigació i Innovació Parc Taulí, Universitat Autònoma de Barcelona, 08193 Sabadell, Spain; 9Respiratory Medicine Department, Hospital Son Espases—Instituto de Investigación Sanitaria de Palma (IdISBa), Universitat de les Illes Balears, 07120 Palma de Mallorca, Spain

**Keywords:** COPD, mortality, prognosis, proteomic fingerprint, immunity, hemostasis

## Abstract

Chronic Obstructive Pulmonary Disease (COPD) is the third leading cause of global mortality. Despite clinical predictors (age, severity, comorbidities, etc.) being established, proteomics offers comprehensive biological profiling to obtain deeper insights into COPD pathophysiology and survival prognoses. This pilot study aimed to identify proteomic footprints that could be potentially useful in predicting mortality in stable COPD patients. Plasma samples from 40 patients were subjected to both blind (liquid chromatography–mass spectrometry) and hypothesis-driven (multiplex immunoassays) proteomic analyses supported by artificial intelligence (AI) before a 4-year clinical follow-up. Among the 34 patients whose survival status was confirmed (mean age 69 ± 9 years, 29.5% women, FEV_1_ 42 ± 15.3% ref.), 32% were dead in the fourth year. The analysis identified 363 proteins/peptides, with 31 showing significant differences between the survivors and non-survivors. These proteins predominantly belonged to different aspects of the immune response (12 proteins), hemostasis (9), and proinflammatory cytokines (5). The predictive modeling achieved excellent accuracy for mortality (90%) but a weaker performance for days of survival (Q^2^ 0.18), improving mildly with AI-mediated blind selection of proteins (accuracy of 95%, Q^2^ of 0.52). Further stratification by protein groups highlighted the predictive value for mortality of either hemostasis or pro-inflammatory markers alone (accuracies of 95 and 89%, respectively). Therefore, stable COPD patients’ proteomic footprints can effectively forecast 4-year mortality, emphasizing the role of inflammatory, immune, and cardiovascular events. Future applications may enhance the prognostic precision and guide preventive interventions.

## 1. Introduction

Chronic obstructive pulmonary disease (COPD) is a very prevalent disorder, and is the third leading cause of death worldwide [1], which also negatively impacts not only health systems (costs, overload) but socio-economic outcomes in general [2]. Therefore, it is important to establish good predictors for the more relevant events occurring in patients’ lives: exacerbations, hospitalizations, disability, and mortality [3]. The rate of the latter depends on the characteristics of the patients but also on the specific health system, oscillating from 25 to 40% at 4–5 years in individuals with a stable clinical condition to around 50% at 2 years in hospitalized patients [4,5,6,7,8,9]. Most of the long-term deaths are attributable to respiratory causes, with lower percentages attributed to cardiovascular/cerebrovascular events or cancer [4,10]. Many different clinical predictors of mortality, including age, obesity, dyspnea, hypoxemia, anemia, exercise capacity, disease severity, nutritional status, comorbidities, and multidimensional indices such as BODE, have already been described [3,4,11,12,13,14,15,16,17,18,19]. There have also been some attempts to obtain blood markers that can predict patients’ death, for example, neutrophil, eosinophil, or lymphocyte counts, and the levels of proatrial natriuretic peptide (MRproANP), copeptin, endotrophin, von Willebrand factor (VWF), and D-dimer, among others [20,21,22,23,24]. Many of these studies were, however, retrospective. Moreover, it is worth noting that a trend toward better prognoses (disease progression, exacerbations, and mortality) has been observed in the last few years. This phenomenon may be due to recent changes in treatment [3,11,19,25,26,27] and involves the need for new studies on the current predictors of mortality.

We hypothesized that the plasma proteomic profile of stable COPD patients, which is a surrogate reflecting their pathophysiological abnormalities, could be useful in determining their survival prognosis. Therefore, the objective of this pilot study was to identify proteomic markers and specific multiprotein signatures that could be useful in predicting long-term mortality, identifying the related pathways and contributing to our knowledge on the mechanisms linked to a poor prognosis for COPD patients.

## 2. Materials and Methods

### 2.1. Study Design and Ethics

This prospective study was performed with patients from the BIOMEPOC cohort (269 patients and 83 healthy subjects), whose details have been published elsewhere [28]. The investigation was designed and conducted according to the Helsinki Declaration and all procedures were approved by our institutional ethics committee (ref. 2014/5895/I). Moreover, all participants signed the corresponding written informed consent form.

### 2.2. Study Population

For the current pilot analysis, a subset from the BIOMEPOC cohort comprising randomly chosen stable COPD patients was included and followed for 4 years. All patients were Caucasians from the European Mediterranean area. The diagnosis of COPD was based on smoking history and a post-bronchodilator spirometry ratio (FEV_1_/FVC) < 0.7 [2,28], whilst clinical stability was defined as the absence of any significant clinical changes including acute exacerbations in the three months preceding the beginning of the study. The exclusion criteria included all other respiratory or chronic inflammatory conditions. Clinical data were recorded at the beginning of the study.

### 2.3. Biological Sample Obtention

Blood samples were collected by venipuncture at the onset of the study, which were immediately placed in K3-EDTA tubes for plasma analyses. These tubes were centrifuged at 1500 r.p.m. at 4° C for 15 min, and the supernatants were moved to new tubes and stored at −80 °C until the proteomic analyses were performed. The latter were performed through two different but complementary techniques.

### 2.4. Liquid Chromatography–Tandem Mass Spectrometry (LC–MS/MS)

Details of the sample preparation, instrument parameters, and protein identification process have been published elsewhere [29,30]. Briefly, the proteins were cleaved with trypsin (1:100 *w*:*w*, 8 h) (Promega, Madison, WI, USA) and endoproteinase LysC (1:100 *w*:*w*, overnight) (Merck, Wako, Neuss, Germany). The reactions were carried out at 37 °C and the peptides were desalted in C18 Hypersep columns (Thermo Fisher Scientific, Waltham, MA, USA). The peptides were analyzed by LC–MS/MS using an LTQ-Orbitrap Fusion Lumos mass spectrometer linked to an EASY nLC 1000 nanocapillary and microflow system (Thermo Fisher Scientific). Digested samples from each patient were tested in duplicate, and label-free quantification (LFQ) and database inquiries were performed using the MaxQuant LFQ software (version 1.6.1.0, Max Planck Institute for Biochemistry, Martinsried, Germany), the Andromeda algorithm, and the *H. sapiens* database (SwissProt, Geneva, Switzerland 2018). Protein amounts were approximated using MaxQuant LFQ values.

### 2.5. Immune-Based Multiplexing

The results obtained with the previous blind method were expanded by using a complementary hypothesis-driven approach focused on soluble inflammatory markers, including cytokines, chemokines, growth factors, and acute-phase proteins. This was achieved using 3 multiplex bead-based immunoassays and a Bio-Plex 200 array system (Bio-Rad, Hercules, CA, USA). Sixty-three distinct plasma proteins were assessed with this procedure following the manufacturer’s recommendations. The validation of randomly chosen proteins from the immunoassay results was conducted using ELISA in the standard way. The details have been described previously [29].

### 2.6. Data Analysis

#### 2.6.1. Calculation of the Sample Size

The minimum sample size to be included in the present pilot study was based on previously published studies and calculated using the GRANMO software (version 8.0) [29,30,31], assuming a loss of 10% of patients in the follow-up period.

#### 2.6.2. Descriptive Statistics and Comparisons between Groups

Clinical variables were tested for normal distribution using the Kolmogorov–Smirnov test. Since all of them showed a normal distribution, their results are presented as mean ± SD, and comparisons between the survivor and non-survivor groups were performed using the unpaired *t*-test.

Regarding proteomic data, the results from both techniques were merged into a single dataset and protein values were log_2_ transformed to diminish the skewness of their distributions. These data are presented as mean ± SD or median and quartiles depending on the presence or absence of normality in their distributions. Proteomic data comparisons were analyzed using complementary quantitative and qualitative approaches (see the flow chart diagram in Appendix A).

A “95% minimum imputation” of missing values was performed [30,32] and quantitative comparisons between the survivor and non-survivor groups were made for the proteins detected in at least 80% of patients from each group. Independent sample t- or Mann–Whitney tests were employed for comparisons between groups, as appropriate. The protein log-base-2-fold change (log2FC) value was determined by subtracting the average log_2_ values of the proteins from each comparison of the survivor and non-survivor groups. The remaining proteins were included in a qualitative analysis to identify the ones that were consistently present in all/almost all of the patients from one of the two groups but were absent in the other (P-A Proteins). This analysis was performed using the Barnard test. The proteins showing differences between the two groups in any of the statistical approaches (identified as differentially abundant proteins, DAPs) were used for the network analysis.

The SPSS v.28 software (SPSS Inc.; Chicago, IL, USA) was used for the descriptive analysis of the clinical data, whereas Python 3.8.3, in a Jupyter notebook v6.0.3 associated with an Anaconda environment v4.9.1 (NumPy v1.18.5, Pandas v1.0.5, SciPy v1.5.0, and Statsmodels v0.11.1 Python modules) was used for the proteomic data.

### 2.7. Functional Classification of Proteins and Network Analysis

The proteins were categorized using Reactome v.88 and UniProtKB v.2024_03 into 7 categories (lipid metabolism-related, hemostasis, inflammatory mediators, complement system, adaptive immunity, other immune-related proteins, and “orphans”). The resultant classification was manually refined in certain cases. Detailed descriptions are available in the Appendix A.

To further assess the biological relevance of DAPs, protein–protein interactions (PPIs) were explored through the STRING digital platform (STRING Consortium, Kearney, NE, USA), using a significance score of ≥0.4. Interactions were visualized by generating network representations through Cytoscape (Cytoscape Consortium, San Diego, CA, USA). The proteins that are not included in the STRING database were manually added. Further details can be found in a previous article from our group [29].

### 2.8. Generation of Predictive Models

Predictive models of both mortality (categorical) and days of survival (continuous) were developed using AI–machine learning techniques: Random Forest (RF), Partial Least Squares Regression (PLSR), and K-Nearest Neighbors (KNN). These models allow for a deep understanding of the characteristics of the predictions and their level of reliability. Two different approaches were chosen to generate the models, which were assisted by AI. First, only quantitative and qualitative DAPs were included to build the models, whereas in the second step, AI was allowed to freely choose the proteins that would build the best predictive model. For this second step, although the set was constituted by the qualitative data of all proteins, only the quantitative data from the proteins with 100% coverage in all patients were included. This was chosen to counterbalance the potential limitations of individual imputations, improving the consistency of our model [30].

The specialized software named Flame, an open and free tool developed in our center for predictive modeling (version 1.2.2; https://phi.upf.edu/phi/index.php/software/flame/) [33], was used for the analysis due to its ease of use as well as its robust and replicable results.

*Initial fitting*. The data were preprocessed using standard scaling. The All-KNN method was used as a sub/oversampling strategy to correct class imbalances (death and survival) for continuous modeling (days of survival), and the Instance Hardness Threshold was used for categorical modeling (mortality at 4 years). In all cases, the Random Forest (RF) method was used in a conformal variant with the confidence level set at 0.8, coupled with K-best feature selection using the “auto” feature number selection criterion. Two hundred trees were used in the RF model, and their class weights were considered balanced because of the pre-processing procedure. RF-max features were calculated through the square array of the previously selected features. A minimum of 2 samples were required for RF-split. An internal method with out-of-the-bag samples was used to estimate the generalization accuracy. A random seed of 46 was used for reproducibility. A conformal analysis was conducted using Aggregated Conformal Predictors (ACPs), with model normalization achieved using a KNN method that included the 15 nearest neighbors. These ACPs aggregated conformal predictions derived from 10 models, and a bootstrap sampler strategy was employed to select calibration sets. Median aggregation was employed to combine the resulting *p*-values.

*Internal validation*. The obtained models were validated using a 5-fold cross-validation approach, with a specific permutation importance approach being developed to discern the significance of each molecule within each one of the models [33].

*Model evaluation*. Based on coincidences or discrepancies between the predicted and the real survival situation, a confusional matrix was generated with the latter, and sensibility (SE), specificity (SP), Predictive Positive Value (PPV), Predictive Negative Value (PNV), Overall Detection Accuracy (Acc), and the Matthews correlation coefficient (MCC, commonly known as the phi coefficient, φ or rφ) were calculated. Both our own death rate and the mortality data provided by a large study (close to 9000 patients) from the COPD Cohorts Collaborative International Assessment (3CIA) (Appendix C) [34] were used to calculate PNV, PPV, and Acc.

## 3. Results

### 3.1. General Characteristics of the Patients

The survival outcome at the end of the 4-year follow-up was impossible to confirm in six COPD patients. The remaining 34 were mainly in the seventh and eighth decades of life (mean age: 69 ± 9 years old), 29.5% were women, and all were current or former smokers. The severity of the disease was either mild–moderate (26.5%) or severe–very severe (73.5%), with several exacerbations in the previous year, mostly ranging in number from 0 to 3. The prescribed treatments included long-acting bronchodilators (95%), inhaled or oral steroids (41% and 3%, respectively), regularly scheduled antibiotics (8%), platelet antiaggregants (26%), anticoagulants (11%), and home oxygen therapy (6%). The main comorbidities were cardiovascular disorders (57.8%, including ischemic heart disease, 12.5%; cerebrovascular disease, 7.9%; and peripheral vascular disease, 4.8%), sleep apnea syndrome (25.4%), and diabetes (14.3%), which resulted in a Charlson index of 3.6 ± 1.6. The mortality rates were 6, 18, 24, and 32% at the first, second, third, and fourth year of follow-up, respectively. The main causes of death were respiratory and cardiovascular events (35% and 27%, respectively). Table 1 shows the main clinical differences between 4-year survivors and non-survivors. As expected, the proportion of frequent exacerbators who died in the follow-up period was much higher than that of non-frequent exacerbators (50% vs. 20%, respectively).

### 3.2. Proteomic Profile

From a total of 363 proteins/peptides identified through either LC-MS or multiplexing (listed in Appendix A), the data for 208 were at least 80% valid in both groups and were included in the quantitative comparisons between the survivors and non-survivors (Appendix A). Fifteen of them showed significant differences, and their categorization revealed that they were mostly associated with four different biological processes: hemostasis (n = 5), proinflammatory processes (cytokines and chemokines, n = 3), the complement system (n = 3), and other immunological pathways (n = 2, with one of them being an immunoglobulin fraction). The two proteins that were not grouped into any of the categories were called “orphans” (Table 2, Appendix A)

Moreover, sixteen more proteins/peptides with less than 80% valid values were significantly more present in one of the groups, and are or are involved in hemostasis (n = 4), proinflammatory mediators (n = 2), the complement system (n = 1), other immunological pathways (n = 6, with three of them being immunoglobulin fractions), and “orphan” proteins (n = 3) (Table 3, Appendix A).

Globally, in the joint analysis from both statistical approaches, 31 proteins showed statistically significant differences. A protein–protein interaction network was elaborated based on the STRING database and our functional classification groups (Figure 1).

In this joint analysis, the hemostasis group comprised nine proteins. The abundances of coagulation factors II (F2, prothrombin), X (F10, thrombokinase), and XII (F12), the subunit B of platelet-derived growth factor (PDGFB), and plasminogen (PLG, including PLGLA and PLGB1) were lower or mostly absent in the non-survivor patients when compared with the survivors. In contrast, the abundances of the prothrombotic factors talin-1 (TLN1) and protein Z (PROZ), as well as the antihemostatics α-2 macroglobulin (A2M) and tyrosine-protein phosphatase non-receptor type 11 (PTPN11), were higher or mostly present in the patients who did not survive.

The group of inflammatory mediators was composed of five proteins/peptides. The abundances of IL-1β and CCL17 were lower or absent and that of CXCL9 was higher in the patients who died; CXCL5 and CSF2 (also known as GM-CSF, granulocyte–macrophage colony-stimulating factor) were mostly absent in this group.

The immune-related pathway group contained a total of 12 proteins, which can be further subdivided into three different subgroups: those related to the complement system (innate immunity), the adaptive immune response, and ‘other immune pathways’. Four proteins were associated with the complement system: the abundances of subunits A and C of the subcomponent C1q (C1QA and C1QC, respectively) were higher in the patients who were deceased at the 4-year follow-up, whereas the abundances of factor H (CFH) and its inhibitor properdin (CFP) were lower or mostly absent. The adaptive immunity subgroup comprised immunoglobulin light chain λ variable 3-25 (IGLV3-25) and light chain Ϗ variable 6-21 (IGKV6-21), which were mostly absent in patients who died, as well as immunoglobulin light chain λ variable 3-10 (IGLV3-10) and heavy chain variable 1-5 (IGHV2-5), whose abundances were higher in this group. The proteins included in the ‘other immunological pathways’ group were peptidoglycan recognition protein 2 (PGLYRP2, an enzyme that hydrolyzes a component of bacterial cell walls) and attractin (ATRN, also involved in the inflammatory response), whose abundances were lower or mostly absent in non-survivors, whereas solute carriers from family 2 (SLC2A, including SLC2A3 and SLC2A14 members) and the engulfment protein GULP adaptor 1 (GULP1) were mainly present in non-survivors at 4 years.

Finally, the five differentially abundant proteins that were not included in the previous groups (“orphan” proteins) are the antioxidant glutathione peroxidase 3 (GPX3), myosin light chain 6 (including MYL6 and MYL6B, which are motor proteins expressed in different tissues), and an insulin-like growth factor-binding protein (IGFALS, a facilitator of the actions of insulin-like growth factor); all of their abundances were lower or absent in the non-survivor patients. Meanwhile, the abundances of an olfactory receptor from family 5, subfamily M (member 11 (OR5M11), which is involved in the transduction of odorant signals) and FAM71F1 (regulates acrosome formation) were higher in this subpopulation.

The details of the results for patients taking either platelet antiaggregants or anticoagulants were excluded from the analysis but can be found in the Appendix A. Briefly, some of the DAPs in the hemostasis group remained significant while the others did not reach statistical significance.

### 3.3. Prediction of Death and Days of Survival (Table 4 and Table 5)

#### 3.3.1. Conventional Approach

In a first step, two models were developed to predict mortality: one is categorical, based on death/survival at the 4th year, and the other is ‘continuous’, based on days of survival. For this purpose, the 31 DAPs identified in the previous phases of the analysis were initially used.

The categorical model gave excellent results in both the initial fitting (accuracy of 100%) and the subsequent internal validation, based on either the reported literature mortality (47%, Appendix C) or deaths in our cohort (32%) (accuracies of 90 and 93%, respectively; Table 4a).

**Table 4 cells-13-01351-t004:** Summary of the results for categorical models (mortality).

	Fitting			Prediction (Internal Validation)						
Model Name	Prot	Se/Sp/Acc/MCC	Cov	Se	Sp	MCC	Cov	PPV (Rep|Our)	NPV (Rep|Our)	Acc (Rep|Our)
(a) *Conventional*	31	1.00	1.00	0.78	1.00	0.79	0.77	1.00|1.00	0.84|0.91	0.90|0.93
(b) AI free choice	10	1.00	1.00	0.89	1.00	0.89	0.82	1.00|1.00	0.91|0.95	0.95|0.96
- Hemostasis group only	10	1.00	1.00	1.00	0.90	0.88	0.73	0.90|0.82	1.00|1.00	0.95|0.93
- Cytokines only	10	1.00	0.68	0.80	1.00	0.80	0.53	0.82|0.70	1.00|1.00	0.89|0.86

Abbreviations: Prot, proteins; Se, sensitivity; Sp, specificity; MCC, Matthews correlation coefficient; rep, based on reported mortality rate (47%, Appendix C); ours, based in our mortality rate (32%); Cov, coverage; PPV, positive predicted value; NPV, negative predicted value; Acc, accuracy.

**Table 5 cells-13-01351-t005:** Summary of results for continuous models (days of survival).

		Fitting		Prediction	
Model Name	Proteins	R^2^	Conformal Accuracy	Q^2^	Conformal Accuracy
(a) *Conventional*	31	0.64	1.00	0.18	0.95
(b) AI free choice	10	0.81	1.00	0.52	0.95
- Hemostasis group only	10	0.64	1.00	0.25	0.91
- Cytokines only	10	0.71	1.00	0.36	0.95

The continuous model showed an initial R^2^ of 0.64 (usually considered as “acceptable” [35]) between the model prediction and the actual days of survival. However, the R^2^ dropped to just a Q^2^ of 0.18 (considered “weak”) in the internal validation (Table 5a).

#### 3.3.2. AI Free Choice of Proteins

In the second step to improve the results of our initial prediction, we allowed the AI modeling to freely chose the 10 proteins that best contributed to generating the most accurate prediction model.

Indeed, the categorical model (i.e., death vs. survival at 4 years, Table 4b) obtained was able to better predict mortality at 4 years, not only in the fitting but also in the internal validation steps, using either literature data or our own mortality data (accuracies of 100, 95, and 93%, respectively). The ten proteins used in this model belonged to the following functional groups: hemostasis [n = 4; kininogen (KNG1), PLG, platelet factor 4 (PF4), and thrombospondin 1 (TYHBS1)], innate and adaptive immunity pathways (n = 4; CFH from the complement system, as well as IGKV6-21, IGLV6-57, and PGLYRP2), and ‘orphan’ proteins (n = 2, including MYL6 and a nutrition-related protein known as afamin) (Figure 2a).

The continuous model (i.e., days of survival) initially disclosed an R^2^ of 0.81 (strong) between the predicted and actual days of survival, decreasing to 0.52 (‘acceptable’) in the validation step (Table 5b). The 10 proteins/peptides present in this latter model belonged to the proinflammatory mediator (n = 3; IL-1β, CXCL5, and CLL25), hemostasis [n = 3; F10, fibrinogen (FG), and α-1-acid glycoprotein 1 (ORM1, which interacts with PAI-1 favoring coagulation, although it can also be considered as part of the inflammatory response)], immunological pathway (n = 1; SLC2A), and ‘orphan’ [n = 3, including the component of structural muscle proteins MYL6, as well as fibrocystin (PKHD1) and mediator of DNA damage checkpoint protein 1 (MDC1)] groups (Figure 2b).

Figure 3 shows a visualization of the interactions between the proteins identified as predictive in the initial and more conventional analysis and those freely chosen by the AI for both categorical and continuous predictions (death and days of survival, respectively).

When the models were built with only one of the five biological groups chosen for the analysis, those that included hemostasis or proinflammatory mediators produced the best results in both analyses, in close accordance with their relative importance in the models (Appendix A).

Indeed, the hemostasis group reached 100% accuracy in the fitting step, as well as accuracies of 95 and 93% in the validations (calculated using, as always, the reported and our own mortality rates, respectively) in the categorical prediction (Table 4b). However, the 0.61 R^2^ value obtained in the fitting of the continuous analysis fell to a Q^2^ of 0.25 in the validation step (Table 5b).

Similarly, the group of proinflammatory mediators (cytokines) produced excellent results in the categorical prediction (accuracies of 100% for fitting, and 89 and 86% in both validations) (Table 4b) and the fitting of the continuous one (R^2^ of 0.71, Table 5b). However, the validation of the latter decreased to a ‘weak’ classification (Q^2^ of 0.36, Table 5b). Aggregations of the same five biological groups into sets of two did not improve these results.

Details on the models obtained for patients taking antihemostatic drugs were omitted but can be found in the Appendix A. In summary, when the patients taking platelet antiaggregants were excluded, the models did not vary substantially. Moreover, when the patients under anticoagulant therapy were discarded, the AI chose six instead of four proteins from the hemostasis group, improving the accuracy of the model for categorical prediction. However, the results for the continuous prediction were modest.

## 4. Discussion

To our knowledge, this is the first study to analyze the predictive ability of blood proteomics on long-term mortality (4 years) in stable COPD patients through the combination of two complementary conceptual approaches (non-hypothesis-biased and hypothesis-driven analyses). Although this was a pilot study with a relatively low number of patients, our results indicate that the plasma proteomic profile can be used for death prognosis purposes. The markers obtained during clinical stability and linked to proinflammatory status, as well as to hemostasis and different components of the immune response, appear to be the main determinants of the long-term death rate.

### 4.1. Previous Studies

The mortality rate of the present pilot study is similar to that reported in most of the previous studies carried out in stable COPD patients from different developed countries [4,7,10,36,37,38]. Previous investigations on factors associated with death in COPD patients have mainly assessed clinical variables, identifying predictors such as comorbidities, frailty, functional capacity, physical activity, nutritional status, dynamic hyperinflation, and exertional desaturation, among others [39,40,41,42,43,44,45]. Interestingly, however, and in close accordance with our proteomic results, respiratory events of infectious origin and cardiovascular disorders, together with cancer, clearly stand out among the comorbidities closely associated with COPD mortality [46,47,48,49,50,51,52]. Fewer studies have focused on prognostic blood markers, and normally, they have been restricted to those obtained in routine tests. This is the case for percentages of neutrophils and lymphocytes, and fibrinogen, hemoglobin, D-dimer, and oxyhemoglobin concentrations [23,53,54,55,56,57]. Other authors have investigated specific markers that are not currently tested in COPD patients, such as IL-6, cholinesterase, copeptin, MRproANP, p-calprotectin, and soluble suppression of tumorigenicity 2 (sST2) [20,58,59,60,61]. However, it seems increasingly clear that only aggregations of various protein biomarkers can identify a signature with a clear predictive value for COPD outcomes. This is the case, for instance, for the combination of CC16, SRAGE, fibrinogen, CRP, and SP-D proposed by Zemans et al. to predict patient mortality [62], or the one introduced by Agustí et al. based on white blood cell (WBC) counts and C-reactive protein (CRP), IL-6, IL-8, fibrinogen, and TNF-α levels [63]. Nevertheless, all these were target-driven studies, addressing specific molecules that were already considered potential candidates for being predictive of negative outcomes. Only a few studies have attempted a wider biological approach to clinical COPD outcomes using large-scale techniques [64,65]. Unfortunately, most of them did not address mortality. This is the case for Zhang et al., who used a combination of metabolomics and proteomics to identify severity biomarkers for the disease [66]. These authors found that different proteins associated with hemostasis and/or endothelial function were dysregulated in COPD, although only cadherin (5CDH5) was a good marker of an advanced disease. From those that investigated markers of future death in stable COPD patients, Gregory et al. clearly stand out. These authors performed a cluster analysis using transcriptomics and proteomics in a large cohort of COPD patients (COPD Gene) [67], obtaining three groups of predictive proteins: those linked to innate immunity, mitochondrial function and cytoskeleton rearrangements-fatty acid metabolism.

Certainly, some complementary studies have also investigated predictors of mortality during acute COPD exacerbations. However, this is a totally different subject and is more focused on short- to medium-term prognoses in already unstable patients. As it is also the case in stable patients, the clinical findings have been prominently used to establish a survival prognosis [68,69], but some blood biomarkers, such as troponin levels, the eosinophil-to-platelet ratio, the triglyceride–glucose index (TyG), and CRP levels, among others, seem to be useful for this purpose [68,69,70,71].

### 4.2. Interpretation of Novel Findings

#### 4.2.1. Differentially Abundant Proteins

The present investigation revealed that immune-related proteins, including those belonging to the complement system, and proteins associated with hemostasis are the main groups of markers that were differentially abundant between survivors and non-survivors. This was a prominent feature in the results from both the more conventional and the free-choice AI analyses.

Regarding immune-related pathways, we considered three different components; one of these is proteins linked to adaptive immunity, such as immunoglobulins, but our results initially seemed to be partially contradictory. There was an increase or a differential presence of some immunoglobulin fractions (IGLV3-10 and IGHV2-5) in the patients who later died, but other fractions were absent (IGLV3-25 and IGKV6-21). Immunoglobulins are key elements of the humoral response, allowing for the control of infections through opsonization, complement cascade activation (see the following paragraph), and cytotoxicity. A possible interpretation of the present findings may be that the activation of the adaptive immune response in apparently stable patients occurred, but it was partially compromised. This would be consistent with the observations of our and other previous studies in patients with frequent exacerbations [29,30,72], a specific COPD phenotype that (as also happened in the present study) consistently shows an increased mortality rate [73].

Moreover, other immune mechanisms may also have been defective in patients who would die later, as suggested by the absence of ATRN, a known modulator of the immune and inflammatory response, or the lower level of PGLYRP2, which is involved in the destruction of bacterial walls. On the contrary, other immune-related proteins seem to have been activated in those patients who subsequently died. This would have been the case for the presence of either SLC2A, which is involved with lymphocyte signaling and the subsequent activation of the response against infections and tumors [74], or GULP, implicated in the phagocytic elimination of cells and microorganisms that are already inactive, in this COPD group.

The complement system, in turn, showed signs of activation in the first steps of the classical pathway (moderate increases in C1q A and C chains), but a probable restriction of the common final steps of the system, which are especially critical for the alternative pathway, through the inhibition of its C3 self-amplification loop (i.e., the decrease in CFH and the apparent absence of CFP). The complement cascade is an essential part of the defense system since it directly induces bacterial lysis and opsonization, and contributes to improving the action of antibodies and clearing immune complexes [75]. This system is made up of three different pathways that converge in their final steps. In the main one (‘classical’), the complement component C1 can bind to immune complexes or react with different polyanions, CRP, or even components of bacteria walls. The so-called lectin or MB-lectin path has some similarities with the main pathway, but here, the activation of the system occurs when mannose-binding lectin (MBL) recognizes foreign carbohydrates present in bacterial walls, which is totally independent of antibodies. It is worth noting that MBL deficiency has been associated with the pathogenesis of COPD and high serum MBL levels have been associated with increased survival in stable patients [76]. The third path, usually known as the ‘alternative pathway’, is only slightly active in normal conditions but initiated when components of bacterial walls, such as bacterial lipopolysaccharide (LPS), or even immunoglobulins induce a moderate split of C3. However, these three distinct paths converge when the action of a specific convertase splits C3 into C3a and C3b, leading to the generation of the membrane attack complex (MAC, composed of C5b, C6, C7, C8, and C9), which disrupts the membrane of pathogens, leading to their death. Other authors have also shown that some components of the complement system can be altered in COPD patients [66,67], but the present report contributes to clarify some of these abnormalities and their association with mortality. As previously mentioned, our results suggest that although patients who will die in the next few years have a certain activation of the initial steps of the complement system (which would involve the presence of a relatively permanent harmful noxae), and the final response is impaired, even in periods of clinical stability (a factor that may be a determinant in their future defense capacity against infections) [76]. Indeed, this is in close agreement with our previous observations in either stable or exacerbated patients [29,30]. Although other authors have reported that C9 was one of the main proteins associated with an increased risk of death [67], no differences were found for this component of the complement system between our two groups of patients.

Hemostasis was one of the two groups will more proteins associated with death in our COPD patients. This is in close agreement with some previous clinical studies that evidenced a high prevalence of cardiovascular comorbidities in COPD [49,50,51,52,77]. Moreover, factors involved in thrombogenicity that contribute to severe cardiovascular events, such as increases in coagulation factors, platelet activation, and even endothelial dysfunction, have been described in both stable and exacerbated COPD patients [78]. Other studies have specifically highlighted the relevance of thrombotic events in their survival prognosis [49,50,51,52]. Interestingly, however, the hemostasis-linked proteins/peptides identified in the present study seem to indicate an ambivalent situation in deceased patients. On the one hand, there are elements whose presence or increased levels would either favor the generation of thrombi (TLN1 and PROZ) or hinder their resolution (decreases in PLG). It appears, however, that the levels of other proteins would have potential antithrombotic effects (i.e., the increase in A2M and the presence of PTPN11, as well as decreases or an absence of F2, F10, F12, and PDGFB in models from the conventional analysis). Certainly, an alternative explanation for the results of the latter proteins would be that they have been extensively used in the generation of subclinical thrombi [79]. Moreover, other authors have related low levels of some factors related to hemostasis with difficulties in the repair of previously injured airways and lung tissues in COPD patients [80]. Other researchers have also observed dysregulation in these and other prothrombotic factors (such as F11a and tissue factor, TF) in stable COPD patients, suggesting that this may contribute to an elevated risk of cardiovascular complications [66,81,82]. Moreover, Sand et al. and Manon-Jensen et al. reported that high levels of markers of fibrin clot resolution are indicative of a poor survival prognosis in COPD patients [79,83]. Similarly, Ronnow et al. and Langholm et al. described an increase in mortality in patients with high levels of Von Willebrand factor epitopes [24,84].

Regarding inflammatory mediators, the absence or reduced presence of various cytokines and chemokines in the group of patients who died before the 4-year follow-up may have contributed to impaired modulation of the inflammatory response. For example, reduced levels of CCL17 and CXCL5 may have resulted in a decreased leukocyte chemotaxis, and reduced levels of IL-1β and CSF2 may account for the problems in the differentiation and proliferation of various types of leukocytes [85,86]. In contrast, the CXCL9 level was higher in this group than in the survivors. This Th1-type monokine, which is released by neutrophils and induced by interferon-γ, has been described as being elevated in the airways of COPD patients [87], and its blood levels have been reported as being predictive of hospital readmissions [88]. Moreover, multiple studies have shown mild-to-moderate increases of inflammatory mediators in the blood of stable COPD patients, reflecting a low grade of chronic systemic inflammation [89,90,91,92]. Although some previous work described isolated or groups of inflammatory mediators as associated with mortality (as is the case for CRP) [93,94], and others have used microarrays with the same objective [3,95], this is the first article that provides a broader profile of the cytokines and chemokines associated with death in a long-term follow up. Moreover, our set of proinflammatory mediators was one the best protein groups that, alone, was able to predict mortality in COPD patients. This finding is in close agreement with the results published by Agustí et al. in the ECLIPSE cohort [63] and adds new predictive proinflammatory biomarkers.

Finally, the ‘orphan’ proteins/peptides with differential presence between the survivors and non-survivors included an antioxidant (GPX3), an insulin-like growth factor 1 (IGF) facilitator (IGFALS), and a fraction of muscle structural proteins (MYL6), among others. Although purely speculative, the lower level of GPX3 may have contributed to an environment of oxidative stress in patients, while the absence of IGFALS could have resulted in decreased signaling for cell proliferation (including in leukocytes) and, along with the absence of MLY6, may have had an impact on muscle growth. It is well known that a significant number of patients with COPD present a reduction in their muscle mass [96], and this constitutes a factor for a poorer survival prognosis [97]. Although there were no differences in BMI between the survivors and non-survivors in our study, this plasma finding may suggest the presence of a mechanism that does not show a clear clinical expression. Unfortunately, we did not evaluate the body composition at the time of blood extraction or obtain a BMI value closer to the patients’ death.

#### 4.2.2. Prediction of Death and Days of Survival

The models generated by both the more traditional analysis based on DAPs and through the AI free choice of proteins were enormously efficient in predicting deaths at 4 years, although they were much less accurate in predicting the days of survival for each COPD patient. The prediction of death at 4 years using the more conventional statistical analysis was excellent when all the differential proteins meeting the initial study criteria (presence in 80% of patients or being present in only one of the two groups) were analyzed, obtaining the maximum specificity, although the sensitivity was only moderate [98]. However, this latter parameter improved substantially through the use of the AI free choice option.

Moreover, when separately analyzing the four mechanistic groups of proteins separately, it becomes evident that those constituted by inflammatory or hemostasis mediators could generate reasonable approximations for the occurrence of death.

The type of analysis presented here could eventually provide a new tool for predicting mortality or even clinically unexpected deaths [99,100] based only on the pathophysiological mechanisms that may be involved in the prognosis. In this regard, this tool could help us to identify the patients that require special management. Moreover, our results may help in developing future predictive blood tests.

### 4.3. Strengths and Potential Limitations

The complementary approach provided by both the hypothesis-driven and blinded techniques (immune-based assays and mass spectrometry, respectively) stands out as the most powerful point of the present study. Indeed, the immune-based assay approach allowed for the analysis of protein markers known to be associated with pulmonary disease and with a possible impact on its mortality, while the mass spectrometry approach allowed for a very broad screening of unbiased biomarkers, which are less frequently analyzed in other searches. Furthermore, our research group directly obtained and verified all the clinical data and carried out all the different phases of the proteomic study, a characteristic that is very uncommon in most of the previous omic studies on blood biomarkers. Furthermore, our biological tool could be combined with clinical data in the future to increase its predictive power.

Another interesting point is that, in contrast to many previous studies, our sample consisted of around 30% female COPD patients. It is worth noting that a significant increase in the prevalence of COPD in women has been observed worldwide in recent years, and some authors have related this to the changes in global mortality attributed to the disease [101,102]. Our study is therefore more representative of the present COPD population than some of the previous ones [4,19]. This representativity of the results also account for comorbidities, since around 80–90% of COPD patients are associated with at least one additional chronic condition [102].

However, we should recognize that our population sample is relatively small, which is intrinsic to pilot studies but can be considered a potential limitation. Nevertheless, although this restriction may have reduced the power of our analysis, many interesting associations between protein markers and mortality were evidenced. On the other hand, our sample is similar to that of other previous reports using omics to explore biomarkers in COPD [64,66,103,104,105]. We are currently developing a validation approach using multiomics in a much larger sample of patients and with a longer prospective follow-up [106], which might also facilitate the investigation of the impact of co-morbidities and other clinical factors that were not part of the objectives of the present pilot study. Moreover, most of the previous studies on mortality with a higher number of participants were retrospective and/or based solely on clinical variables. In addition to these common clinical or biological variables, our study also identified a wide number of proteins/peptides present in the blood of stable COPD patients who have been prospectively followed for at least 4 years.

It could also be considered that the inclusion of patients on platelet antiaggregant or anticoagulant treatment interferes with the results. However, this situation corresponds to the real world, where a significant percentage of COPD patients shows vascular comorbidities (cardiovascular or neurovascular comorbidities, or involvement of peripheral vessels). Furthermore, our results remain essentially similar regarding the influence of each protein group if such patients are excluded.

Our findings may be subject to variability when new medications or treatment recommendations are introduced. Therefore, ongoing refinement and updates of this biological approximation will be required to achieve real and applicable personalized medicine.

Finally, we should also admit that some of the differentially abundant proteins identified in the present study may have been allocated into more than one biological group. However, we tried to include them in the one that seemed most appropriate for the objectives of the study.

## 5. Conclusions

A proteomic blood signature found in stable COPD patients can help in establishing their long-term (4-year) survival prognosis. The most important elements for establishing this prognosis are proteins/peptides linked to the hemostatic and inflammatory statuses during the stable phase, as well as different elements of the immune response. Furthermore, the present results may help us to better understand the biological mechanisms involved in future deaths in long-term patients.

## Figures and Tables

**Figure 1 cells-13-01351-f001:**
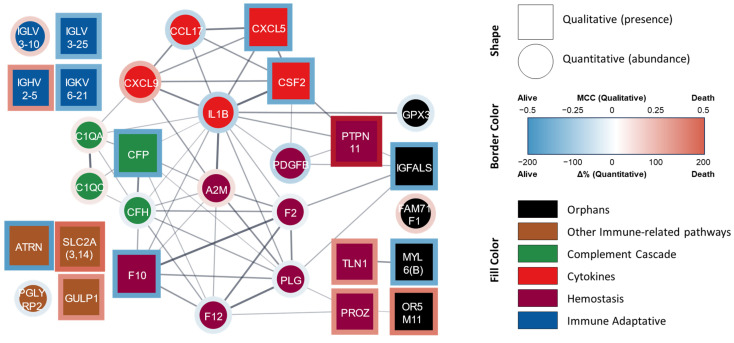
Protein–protein interaction network of proteins significantly related to mortality at four years. STRING-generated network where each node lists the gene symbol or the Ig fraction of the identified proteins. Square nodes represent DAPs identified by the qualitative analysis whereas circle nodes represent the DAPs identified by quantitative analysis. The border color represents the stats value. The fill color represents the assigned functional group: ‘hemostasis’ (purple), ‘cytokine’ (red), ‘complement cascade’ (green), ‘immune adaptive’ (blue), ‘other immune-related pathways’ (brown), and ‘orphan’ (black). A0A075B6K4 (IGLV3-10), A0A0C4DH24 (IGKV6-21), P01717 (IGLV3-25), and P01817 (IGHV2-5) were manually added since they are not included in the STRING database. Abbreviations: Δ%, percent change; MCC, Matthews correlation coefficient (also called the phi coefficient, φ or r_φ_).

**Figure 2 cells-13-01351-f002:**
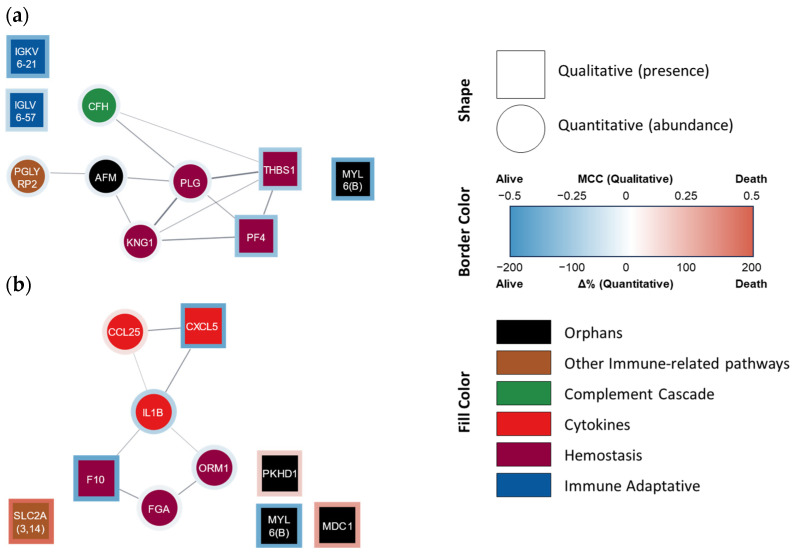
Protein–protein interaction network of proteins that AI selected for (**a**) categorical and (**b**) continuous mortality models. STRING-generated network where each node lists the gene name/Ig fraction of the AI-selected proteins. Square nodes represent qualitative variables, while circle nodes represent quantitative variables. The border color represents the presence/abundance at 4 years in non-survivor (red) and survivor (blue) patients. The fill color represents the assigned functional group: ‘hemostasis’ (purple), ‘cytokine’ (red), ‘complement cascade’ (green), ‘adaptive immunity’ (blue), ‘other immune-related pathways’ (brown), and ‘orphan’ (black). Ig fractions A0A0C4DH24 (IGKV6-21) and P01721 (IGLV6-57) were manually added since they are not included in the STRING database.

**Figure 3 cells-13-01351-f003:**
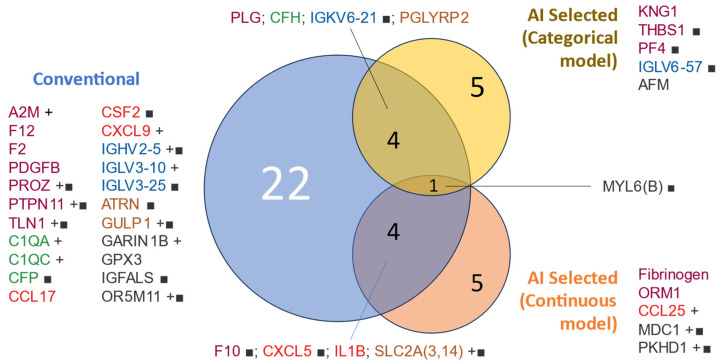
Venn diagram of protein variables used for modeling by selection method. Proteins included in the mortality modeling were selected based on a conventional univariable analysis (*t*-test/Bernard test) or AI selection. The text color represents the assigned functional group: ‘hemostasis’ (purple), ‘cytokine’ (red), ‘complement cascade’ (green), ‘adaptive immunity’ (blue), ‘other immune-related pathways’ (brown), and ‘orphan’ (black). Protein lists are sorted by group and name. ■: qualitative variable (present/absent), +: higher abundance in non-survivors.

**Table 1 cells-13-01351-t001:** Clinical characteristics of the study population.

	COPD 4-Year Survivors (n = 23)	COPD 4-Year Non-Survivors (n = 11)
**General Characteristics**		
Age, year	67 ± 9	72 ± 7
Males, n (% in the group)	15 (65%)	9 (82%)
BMI, kg/m^2^	25.4 ± 6.6	25.0 ± 6.5
**Exacerbator profile**		
FE, n (% in the group)	7 (30%)	7 (64%)
AE last year, n	1.7 ± 1.8	2.4 ± 3.8
**Smoking status**		
Current, n (%)	6 (26%)	4 (36%)
Ex-smoker, n (%)	17 (74%)	7 (64%)
Pack/year smoking	52.2 ± 24.2	54.0 ± 20.6
**Lung Function**		
Post-BD FEV_1_, % pred	42 ± 15	42 ± 15
Post-BD FEV_1_/FVC, % pred	49 ± 12	44 ± 9
DLco, %pred	48 ± 20	44 ± 13
**GOLD Stages**		
I-II, n (% in the group)	5 (22%)	4 (36%)
III-IV, n (% in the group)	18 (78%)	7 (64%)
A-B, n (% in the group)	7 (30%)	2 (18%)
E, n (% in the group)	16 (70%)	9 (82%)
**Conventional Blood Analysis**		
Leucocytes, /µL	8763 ± 2673	8313 ± 2673
Neutrophils, /µL	5627 ± 2333	5795 ± 2302
Eosinophils, /µL	259 ± 240	170 ± 123
CRP, mg/dL	0.8 ± 1.4	1.0 ± 1.1
Fibrinogen, mg/dL	211 ± 57	203 ± 37

Values are shown as mean ± SD or percentage. No significant differences were observed between both COPD groups. Abbreviations: AE, acute exacerbation; BMI, body mass index; COPD, Chronic Obstructive Pulmonary Disease; FE, frequent exacerbator (≥2/year); Post-BD, post-bronchodilation; FEV_1_, forced expiratory volume in the first second; FVC, forced vital capacity; DLco, diffusion capacity for carbon monoxide; pred, predicted; CRP, C reactive protein.

**Table 2 cells-13-01351-t002:** Quantitative differentially abundant proteins (DAPs).

Protein/ Ig Fraction	Protein Name	Functional Classification	%Δ	*p*-Value
A2M	Alpha-2-macroglobulin	Hemostasis	26.105	0.024
F12	Coagulation factor XII	Hemostasis	−27.265	0.038
F2	Prothrombin	Hemostasis	−14.521	0.046
PDGFB	Platelet-derived growth factor subunit B	Hemostasis	−69.182	0.015
PLG	Plasminogen	Hemostasis	−20.748	0.017
C1QA	Complement C1q subcomponent subunit A	Complement cascade	18.952	0.045
C1QC	Complement C1q subcomponent subunit C	Complement cascade	21.426	0.032
CFH	Complement factor H	Complement cascade	−17.151	0.022
CCL17	C-C motif chemokine 17	Cytokine	−63.547	0.035
CXCL9	C-X-C motif chemokine 9	Cytokine	85.719	0.029
IL1B	Interleukin-1 beta	Cytokine	−73.025	0.003
IGLV3-10	Immunoglobulin lambda variable 3-10	Adaptive immunity	53.784	0.046
PGLYRP2	N-acetylmuramoyl-L-alanine amidase	Other immune-related pathways	−25.314	0.018
GARIN1B	Golgi-associated RAB2 interactor protein 1B	Orphan	54.938	0.021
GPX3	Glutathione peroxidase 3	Orphan	−28.710	0.050

Abbreviation: %Δ, percent change.

**Table 3 cells-13-01351-t003:** Qualitative differentially abundant proteins (DAPs).

Protein/ Ig Fraction	Protein Name	Functional Classification	MCC	*p*-Value
F10	Coagulation factor X	Hemostasis	−0.403	0.022
PROZ	Vitamin K-dependent protein Z	Hemostasis	0.357	0.041
PTPN11	Tyrosine-protein phosphatase non-receptor type 11	Hemostasis	0.506	0.004
TLN1	Talin-1	Hemostasis	0.346	0.048
CFP	Properdin	Complement cascade	−0.403	0.022
CSF2	Granulocyte–macrophage colony-stimulating factor	Cytokine	−0.381	0.033
CXCL5	C-X-C motif chemokine 5	Cytokine	−0.403	0.022
IGHV2-5	Immunoglobulin heavy variable 2-5	Adaptive immunity	0.346	0.048
IGKV6-21	Immunoglobulin kappa variable 6-21	Adaptive immunity	−0.358	0.034
IGLV3-25	Immunoglobulin lambda variable 3-25	Adaptive immunity	−0.346	0.048
ATRN	Attractin	Other immune-related pathways	−0.451	0.016
GULP1	PTB domain-containing engulfment adapter protein 1	Other immune-related pathways	0.357	0.041
SLC2A(3,14)	Solute carrier family 2, facilitated glucose transporter member 13 and/or 14	Other immune-related pathways	0.471	0.007
IGFALS	Insulin-like growth factor-binding protein complex acid labile subunit	Orphan	−0.403	0.022
MYL6(B)	Myosin light polypeptide 6 or chain 6b	Orphan	−0.384	0.027
OR5M11	Olfactory receptor 5M11	Orphan	0.403	0.022

Abbreviation: MCC, Matthews correlation coefficient (also called the phi coefficient—φ or r_φ_).

## Data Availability

Data are contained within the article and Appendix A.

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
