# Peer review of "A Pilot Study on Proteomic Predictors of Mortality in Stable COPD"

_cells, 2024, doi:10.3390/cells13161351_

Round 1
Reviewer 1 Report
Comments and Suggestions for Authors
The paper is interesting and well written.
In results, better explanation in 3.2.1 and especially 3.2.2 is needed.
The discussion is a very comprehensive.
Author Response
POINT-BY-POINT Response to the Reviewers
Reviewer 1 Comments
The paper is interesting and well written.
Thank you for your kind review and comments, which allowed us to try to improve the manuscript.
We totally agree with the Reviewer that the Methods and Results could be better and more clearly described.
To this end, we have added two additional subheadings in the Methods section (“Initial fitting” and “Internal validation”) and improved the description of both the steps as well as that of the "model evaluation".
- In results, better explanation in 3.2.1 and especially 3.2.2 is needed.
We appreciate this comment. We have implemented some modifications in the 3.2.1 and 3.2.2 of the Results to improve the clarity and understanding. We hope those changes fulfil the Reviewer’s requirements.
We have also made some modifications in Tables 4 and 5 to facilitate the understanding of the Results as well as adding a legend in Figure 3.
Finally, the Supplementary data document has also been improved to improve understanding.
The discussion is very comprehensive
We thank the reviewer for this compliment.
Reviewer 2 Comments
Thanks for the opportunity to review this article. Overall the article is well written and clear. the authors are able to create a mortality predictor in Stable subjects with COPD using a broad proteomics analysis. I have two comments
Thank you for your review and the positive consideration of our manuscript as well as for the comments and suggestions.
- As the authors describe there are clear features, phenotypes in patients with COPD associated with mortality. exacerbations is one of them. 60% of patients that died had where exacerbators compare to 30% of the group that survive. I assume the clinical data is recorded at day 1, do the authors have a complete set of clinical data? FEV1, co-morbidities, exacerbations, etc. it will be of great interest to see how their proteomics model prediction compares to a clinical data model prediction?
Thanks for those amazing questions. First of all, the reviewer is right. In our study, as we aimed to explore a "very realistic" representation of stable COPD patients we included them randomly from BIOMEPOC cohort database, irrespectively of their ‘frequent’ or ‘infrequent’ exacerbator’ phenotype. Only in the Results have we included the frequency of this characteristic in the survivor and non-survivor groups.
All clinical data and samples were recorded on the recruiting day (we added a clarification sentence in section 2.2), with the obvious exception of the death and days of survival; and yes, we have a complete clinical record of all the patients. Nonetheless, despite agreeing with the reviewer that the comparison between a clinical and a proteomic prediction could be of great interest, this is not one of the objectives of the present pilot study, which focused on the biological abnormalities that can predict mortality. In the near future we will analyse the results in a new study currently being made, where with a much larger cohort of patients we aim to validate the present results and combine multiomic data obtained from blood samples with clinical data
- Most of the pathways describe in the proteomic analysis are probably share but other conditions, heart disease, diabetes, etc. do the authors though about a way to adjut for that. meaning this are probably pathways of COPD patients with co-morbidites not just COPD. are those co-morbidities driving the mortality (probably case) how to put in perspective the proteomic findings?
Again, we agree with the reviewer that other diseases may share some similar pathways described in the manuscript. Nevertheless, to consider different comorbidities is difficult in this pilot study due to the relatively reduced number of patients. Moreover, the study tries to replicate the clinical real-life, where a wide variety of comorbidities are present in COPD patients. More research should be done on that, and we have included this interesting perspective in our next and more extensive project.
I will enjoy a more robust discussion on how to use this data in the current layout of COPD patients or what is the future application that the authors envision for this type of data analysis.
We must thank the reviewer's comment because it allows us to add some paragraphs to the Discussion, detailing why we consider this study valuable in its future applications and future research steps.
In short, we have included two different elements. First of all, these results showed the relevance of haemostatic and inflammatory pathways in the prediction, which drives clinical attention to factors that can contribute to death and therefore, be prevented. Secondly, we consider that our results may provide the basis for future potential blood tests to analytically characterize patients at risk.
We hope that all the elements incorporated now in the Discussion of the new manuscript can fulfil the reviewer's expectations.

Reviewer 2 Report
Comments and Suggestions for Authors
Thanks for the opportunity to review this article. Overall the article is well written and clear. the authors are able to create a mortality predictor in Stable subjects with COPD using a broad proteomics analysis. I have to comments
1.- As the authors describe there are clear features, phenotypes in patients with COPD associated with mortality. exacerbations is one of them. 60% of patients that died had where exacerbators compare to 30% of the group that survive. I assume the clinical data is recorded at day 1, do the authors have a complete set of clinical data? FEV1, comorbidities, exacerbations, etc. it will be of great interest to see how their proteomics model prediction compares to a clinical data model prediction?
2.- Most of the pathways describe in the proteomic analysis are probably share but other conditions, heart disease, diabetes, etc. do the authors though about a way to adjuts for that. meaning this are probably pathways of COPD patients with co-morbidites not just COPD. are those co-morbidities driving the mortalilty (probably case) how to put in perspective the proteomic findings ? i will enjoy a more robust discussion on how to use this data in the current layout of COPD patients or what is the future application that the authors envision for this type of data analysis.
Author Response
POINT-BY-POINT Response to the Reviewers
Reviewer 1 Comments
The paper is interesting and well written.
Thank you for your kind review and comments, which allowed us to try to improve the manuscript.
We totally agree with the Reviewer that the Methods and Results could be better and more clearly described.
To this end, we have added two additional subheadings in the Methods section (“Initial fitting” and “Internal validation”) and improved the description of both the steps as well as that of the "model evaluation".
- In results, better explanation in 3.2.1 and especially 3.2.2 is needed.
We appreciate this comment. We have implemented some modifications in the 3.2.1 and 3.2.2 of the Results to improve the clarity and understanding. We hope those changes fulfil the Reviewer’s requirements.
We have also made some modifications in Tables 4 and 5 to facilitate the understanding of the Results as well as adding a legend in Figure 3.
Finally, the Supplementary data document has also been improved to improve understanding.
The discussion is very comprehensive
We thank the reviewer for this compliment.
Reviewer 2 Comments
Thanks for the opportunity to review this article. Overall the article is well written and clear. the authors are able to create a mortality predictor in Stable subjects with COPD using a broad proteomics analysis. I have to comments
Thank you for your review and the positive consideration of our manuscript as well as for the comments and suggestions.
- As the authors describe there are clear features, phenotypes in patients with COPD associated with mortality. exacerbations is one of them. 60% of patients that died had where exacerbators compare to 30% of the group that survive. I assume the clinical data is recorded at day 1, do the authors have a complete set of clinical data? FEV1, co-morbidities, exacerbations, etc. it will be of great interest to see how their proteomics model prediction compares to a clinical data model prediction?
Thanks for those amazing questions. First of all, the reviewer is right. In our study, as we aimed to explore a "very realistic" representation of stable COPD patients we included them randomly from BIOMEPOC cohort database, irrespectively of their ‘frequent’ or ‘infrequent’ exacerbator’ phenotype. Only in the Results have we included the frequency of this characteristic in the survivor and non-survivor groups.
All clinical data and samples were recorded on the recruiting day (we added a clarification sentence in section 2.2), with the obvious exception of the death and days of survival; and yes, we have a complete clinical record of all the patients. Nonetheless, despite agreeing with the reviewer that the comparison between a clinical and a proteomic prediction could be of great interest, this is not one of the objectives of the present pilot study, which focused on the biological abnormalities that can predict mortality. In the near future we will analyse the results in a new study currently being made, where with a much larger cohort of patients we aim to validate the present results and combine multiomic data obtained from blood samples with clinical data
- Most of the pathways describe in the proteomic analysis are probably share but other conditions, heart disease, diabetes, etc. do the authors though about a way to adjut for that. meaning this are probably pathways of COPD patients with co-morbidites not just COPD. are those co-morbidities driving the mortality (probably case) how to put in perspective the proteomic findings?
Again, we agree with the reviewer that other diseases may share some similar pathways described in the manuscript. Nevertheless, to consider different comorbidities is difficult in this pilot study due to the relatively reduced number of patients. Moreover, the study tries to replicate the clinical real-life, where a wide variety of comorbidities are present in COPD patients. More research should be done on that, and we have included this interesting perspective in our next and more extensive project.
I will enjoy a more robust discussion on how to use this data in the current layout of COPD patients or what is the future application that the authors envision for this type of data analysis.
We must thank the reviewer's comment because it allows us to add some paragraphs to the Discussion, detailing why we consider this study valuable in its future applications and future research steps.
In short, we have included two different elements. First of all, these results showed the relevance of haemostatic and inflammatory pathways in the prediction, which drives clinical attention to factors that can contribute to death and therefore, be prevented. Secondly, we consider that our results may provide the basis for future potential blood tests to analytically characterize patients at risk.
We hope that all the elements incorporated now in the Discussion of the new manuscript can fulfil the reviewer's expectations.
